# Association between local food policy council coverage and longitudinal household food insufficiency during COVID-19, stratified by race, ethnicity, and income

Larissa Calancie[1]*, Yongyi Pan[1], Karen Bassarab[2], Kristen Cooksey Stowers[3], Anne Palmer[2], Misha Eliasziw[1,4]

1 Friedman School of Nutrition Science and Policy, Tufts University, Boston, Massachusetts, United States of America, 2 Johns Hopkins Center for a Livable Future, Baltimore, Maryland, United States of America, 3 Department of Allied Health Science and Rudd Center for Food Policy and Health, University of Connecticut, Hartford, Connecticut, United States of America, 4 Tufts University School of Medicine, Boston, Massachusetts, United States of America

* Larissa.calancie@tufts.edu

## Abstract

Many local food policy councils (FPCs) worked to increase food access during the COVID-19 pandemic. Our objective was to determine whether households living in states with higher FPC coverage were less likely to experience food insufficiency during COVID-19 compared to households in states with lower local FPC coverage, and to analyze associations by race, ethnicity, and household income. We used a modified Poisson regression approach to estimate the prevalence of household food insufficiency in states with high and low FPC coverage as of 2020, adjusting for age and gender of the survey respondent, and percent of the state's population living in a rural area (N = 1,909,647). Longitudinal food insufficiency was measured via the US Census Household Pulse Survey (May 2020 – May 2023). Lower income households in states with low FPC coverage were more likely to experience food insufficiency during the pandemic than households in states with high FPC coverage (food insufficiency prevalence ratio: 1.05, 95% CI 1.04–1.07, p < 0.001). Lower FPC coverage was associated with significantly more food insufficiency among lower-income non-Hispanic Black (1.05, 95% CI 1.02–1.09, p = 0.003) and white households (1.02, 95% CI 1.00–1.04, p = 0.01). Presence of FPCs may have been a protective factor against food insufficiency for low-income Black and white households during the COVID-19 pandemic. Local FPCs may have potential for promoting resilience and racial equity within food systems.

**Data availability statement:** All data necessary for replicating this study are publicly available on Harvard's Dataverse. Calancie, Larissa, 2024, "Association Between Local Food Policy Council Coverage and Longitudinal Household Food Insufficiency During COVID-19, Stratified by Race, Ethnicity, and Income", https://doi.org/10.7910/DVN/ZYTO2H, Harvard Dataverse, V1.

**Funding:** L.C. received funding from the Kansas Health Foundation Authors of this publication are members of the Food Policy Council Working Group, part of the Nutrition and Obesity Policy Research and Evaluation Network (NOPREN). NOPREN is supported by Cooperative Agreement Number 5U48DP00498-05, funded by the Centers for Disease Control and Prevention's (CDC) Division of Nutrition, Physical Activity, and Obesity (DNPAO) and Prevention Research Centers Program. The findings and conclusions in this publication are those of the author(s) and do not necessarily represent the official position of the CDC or the U.S. Department of Health and Human Services (DHHS). The funders had no role in study design, data collection and analysis, decision to publish, or preparation of the manuscript.

**Competing interests:** The authors have declared that no competing interests exist.

## Introduction

"The ability to self-organize is the strongest form of system resilience." Donella Meadows, 1999

The COVID-19 pandemic caused major disruptions to labor markets, economics, supply chains, and emergency food supports in the United States (US), which strongly affected whether households had access to enough food [1–4]. According to the U.S. Department of Agriculture, food insecurity and very low food security significantly increased in 2021 and 2022, reversing a downward trend in food insecurity since 2011 [5]. Early in the pandemic (March 2020), a survey of almost 1,500 low-income individuals (<250% of the federal poverty line) distributed via an online crowd-sourcing platform found that 64% were food insecure or marginally food secure [6]. The situation was most severe for racial and ethnic minorities and families with children for similar and policy-specific reasons, such as interruptions to normal food supports like food pantries, undocumented individuals ineligible for certain food access supports like Pandemic Electronic Benefits Transfer (P-EBT), and interruptions to regular school meals [6–8].

Food policy councils (FPCs) and similar cross-sector collaborative groups have the potential to improve the resilience of local food systems to withstand shocks, such as pandemics [9–12]. FPCs are networks of relationships among individuals and organizations representing different sectors of the food system, and those relationships can expand or be activated to facilitate community change [13,14]. Over 300 councils operate in the US, Tribal Nations, and Canada [15]. When individuals and organizations working on aspects of a complex food system create a shared agenda and trust each other, they can work synergistically to achieve an outcome that no single individual or organization can do on their own [16]. Moreover, councils are increasingly emphasizing diversity, inclusivity and representation from individuals with lived experience with food insecurity [17]. When FPCs contain or have strong connections to communities that disproportionately experience food insecurity, they might be better positioned to inform responses that meaningfully benefit marginalized communities [18].

During the COVID-19 pandemic, a report by the Food Policy Networks (FPN) project at the Johns Hopkins Center for a Livable Future titled "Pivoting Policy, Programs, and Partnerships: Food Policy Councils' Responses to the Crisis of 2020" documented how councils bridged the gap between policies and local implementation needs, and aligned actions with their shared agendas for strong, equitable food systems in their communities [15]. For example, the Jefferson County Food Policy Council (CO) launched a food pantry support program utilizing FEMA/CARES Act funds and included local procurement language in its request for proposals. The New Orleans Food Policy Action Council (LA) successfully advocated for the state P-EBT application process not to include social security numbers, to extend the application date, and to translate materials into languages other than English. The Philadelphia Food Policy Advisory Council (PA) collaborated with its Commerce Department to work with local medium-sized grocers and farmers markets to start online purchasing

for SNAP/EBT recipients. The Pasco County Food Policy Advisory Council (FPAC) (FL) hosted meetings where a school system nutritionist discussed shared concerns about feeding students due to COVID-19 and USDA rules. As a result of the feedback received, the school district policy was changed to allow the district to serve students not attending a brick-and-mortar school at any location. Columbia Food Policy Committee (SC) facilitated a partnership between The Comet public transportation service and Senior Resources, a Meals on Wheels provider, to transport additional meals to senior clients. Multiple councils coordinated services for individuals who tested positive for COVID-19, such as raising funds for culturally appropriate food for refugee and Latinx community members (The Greater High Point Food Alliance, NC), working with the hospital system and food pantries to ensure patients and families had access to food while quarantining (Adams County Food Policy Task Force, PA), and establishing protocols to provide wrap around services for low-wealth community members (Cultivate Charlottesville Food Justice Net, VA) [15,19]. See the FPN report for more examples and details [15].

The COVID-19 pandemic created a natural experiment to compare longitudinal state-level food insufficiency in states with varying degrees of local FPC council coverage during a crisis. The aims of this study were to i) determine whether households living in states with higher local FPC coverage were less likely to experience food insufficiency during the COVID-19 pandemic compared to households in states with lower local FPC coverage, and ii) whether the association between local FPC coverage and food insufficiency was moderated by race, ethnicity, and income since some FPCs reported focusing their COVID-19 relief efforts on populations groups disproportionally impacted by the pandemic in an effort to reduce food system disparities [15].

## Materials and methods

We analyzed publicly available, de-identified data.

### Outcome variable – food insufficiency

The Household Pulse Survey (HPS) was rapidly developed and deployed by the US Census to provide frequent assessments of how the COVID-19 pandemic was affecting households across the country [20]. An assessment of the validity of the HPS food security-related items reported that the item that asked respondents a specific question about having enough food over a short time horizon (i.e., 7 days) was more accurate than a multi-item scale aiming to assess food security over a longer period [21]. The survey question asked: "In the last 7 days, which of these statements best describes the food eaten in your household?" There were four response options: 1) Enough of the kinds of food (I/we) wanted to eat, 2) Enough, but not always the kinds of food (I/we) wanted to eat, 3) Sometimes not enough to eat, or 4) Often not enough to eat. We considered "sometimes not enough to eat" or "often not enough to eat" responses as food insufficient in our analysis. Food sufficiency is a facet of the more comprehensive concepts of food and nutrition security [22].

### Explanatory variable – local FPC coverage

Local FPC coverage was defined as the percentage of a state population living in an area with an active FPC that operated locally (i.e., a geographic area that was smaller than a state or territory). Thirty-nine states had state-level FPCs; those councils did not count toward "local FPC coverage." Active local FPCs were those that reported to be active in response to the 2021 FPN project's biannual survey of FPCs [19] or were confirmed to be active by the FPN project through direct contact, confirmation from a state-level council, or review of information publicly accessible on the internet. The FPN's biannual survey is the most comprehensive list of FPCs in the country. Three hundred and sixteen US-based councils were contacted to complete the 2021 survey of FPCs. After excluding inactive councils and councils operating at the state or territory-level, responses from 290 active local food councils were included in our analyses.

The percentage of a state population living in an area with an active local council was calculated by dividing the number of state residents living in an area with an active local FPC by the total state population. We used the Census 2021 and the Place Explorer search tool to determine the population within each geographic location where there was an

active local FPC and to extract state population totals [23,24]. We used councils reported geographic focus to determine the population catchment area for each council (e.g., we tabulated the regional population of a council that focused on multiple counties). For example, in Arizona, 2,918,852 people lived in areas with active FPCs (Phoenix – 1,624,569, Pima County – 1,052,030, and Yavapai County – 242,253) and the total state population was 7,276,316. Therefore, the percentage of the population covered by local food councils in Arizona was 40.1% (2,918,852/7,276,316). All spatial processing and map visualizations were conducted in ArcGIS Pro (version 3.5.3, Esri Inc, 2025, Redlands, CA). US state boundary shapefiles were obtained from Esri Inc.

## Individual and household variables

Individual (household participant), household, and state level variables were considered in the analyses. Sub-analyses by major race, ethnic, and income groups were conducted.

Individual and household level variables were extracted from HPS survey data. The participant's age was calculated from their birth year. Their gender, race and ethnicity, and education were also extracted from the HPS dataset. Education was categorized into three groups: Less than high school, high school or equivalent, and more than high school. Total household income was categorized into race- and ethnicity-specific tertiles (Lower third, Middle third, and Upper third) as the distribution of total household income varied between the groups: Hispanic, Black non-Hispanic, and Other non-Hispanic households income tertiles were < $35K, $35 – $75K, and > $75K whereas white non-Hispanic household income tertiles were < $50K, $50K – $100K, > $100K and Asian non-Hispanic household tertiles were < $75K, $75K – $150K, > $150K. Household number of children was categorized into 0, 1–2 and 3 or more.

## State variables

We categorized states into four regions: Midwest, Northeast, South, and West, and indicated if a state-level council was present. We assessed the percent of the state's population living in a rural area, the state's median household income, and the percent of eligible individuals within a state participating in the Supplemental Nutrition Assistance Program (SNAP).

## Analytical sample

The Census first administered the HPS in April – May 2020. The frequency of survey changed across the data collection and dissemination cycles (S1 Table). The HPS was collected weekly and then bi-weekly. Phases 3.3 and later maintained the two-week collection periods but shifted to a two-weeks on, two-weeks off collection approach.

The initial sample included 4,412,510 participants aged 17 years or older who participated in Weeks 1–59 of the HPS. We excluded participants who did not have a record of food insufficiency (n = 341,785), record of income (n = 518,227) and participants from the District of Columbia (n = 52,062) as we were interested in FPC coverage at a state-level (i.e., the DC FPC was not considered in this analysis because having a single council representing the entire population in a small geographic area is quite different than any other case in the dataset). These exclusion criteria resulted in 3,500,436 participants.

Data from weekly and biweekly surveys were first aggregated into 39 monthly periods and then further aggregated into 13 quarterly (i.e., 3-month) time periods. Changes in data collection frequency resulted in sample sizes that were 2–4 times larger in the early quarters compared to the later quarters. To preclude estimates of food insufficiency from the early quarters inadvertently having more precision than from the later quarters, the sample size for each quarter was capped at 150,000 participants by taking a random sample of participants in each oversized quarter. Of the 13 quarters included in the analyses, 10 were capped at 150,000 participants and three (May 2022, August 2022, and November 2022) had smaller sample sizes (147,299; 122,719; and 139,629; respectively). As a result, a total of 1,909,647 participants were included in the present analysis.

 

## Statistical analysis

We used a modified Poisson regression approach (i.e., Poisson regression with a robust error variance) to estimate the prevalence of food insufficiency, conduct inferences about the prevalence ratios, and calculate 95% confidence intervals [25,26]. Each ethnic and racial group was analyzed in a separate regression model. Total household income was included in a three-way cross-product regression term, together with quarter and dichotomized council coverage (≤ 15% vs > 15%). Council coverage was dichotomized at 15% to ease the interpretation of the results. It was also the value that split the number of states equally into high and low coverage groups. Potential confounders were statistically assessed and variables that changed prevalence ratios by more than 10% were considered confounders. The final analytical models included the age and gender of the survey respondent, and the percentage of the population living in a rural area. All statistical analyses were carried out using SAS 9.4 (SAS Institute Inc., Cary, NC), and results with p < 0.05 were deemed statistically significant.

## Results

### Sample characteristics

Individual and state level characteristics were similar in both groups of states (Table 1). Almost 60% of participants in both groups were female, over 75% were white, over 85% had more than a high school education and over 90% had 2 or fewer children in their household. While the number of states was equal between the two groups (n = 25), the household sample size was larger in the group with high FPC coverage. More states with low FPC coverage were in the South compared to the high FPC group (10, 40% vs 6, 24% states) (Table 1, Fig 1). The percent of the population living in a rural area was slightly lower in states with low FPC coverage compared to high FPC coverage (30% in low vs. 25% in high coverage states) and median household income was slightly lower ($69,040 vs $73,572). The percent of eligible individuals participating in SNAP was similar in both groups (82.8% in states with low coverage and 83.4% in states with high coverage).

### Association between FPC coverage and food insufficiency overall

Overall, slightly fewer households in states where more than 15% of their population lived in an area with an active FPC reported food insufficiency during the COVID-19 pandemic than households in states with lower FPC coverage, according to multivariate regression modeling. The food insufficiency prevalence ratio between households in states with low FPC coverage compared to states with high coverage between May 2020 and May 2023 was 1.07 (95% CI 1.06–1.08, p < 0.001) (Table 2).

### Association between FPC coverage and food insufficiency by income

Food insufficiency was clearly higher among families in the lower third of their respective income tertiles, with over 15% of households reporting food insufficiency during the pandemic in states with low FPC coverage (Fig 2a). Overall, FPC coverage served as a significant protective factor against food insufficiency among households in the lower income tertile during the pandemic (food insufficiency prevalence ratio of low to high coverage: 1.05, 95% CI 1.04–1.07, p < 0.001) (Table 2). The prevalence of food insufficiency was consistently higher in states with lower FPC coverage among lower income households from May 2021 through May 2023 (Fig 2a). Food insufficiency rose among households in the lower and middle income tertiles from May 2020 to November 2020 and then dropped between November 2020 and May 2021 when federal nutrition supports such as increased SNAP benefits and stimulus checks were implemented [27]. Food insufficiency was very low (≤5%) among households in the middle income tertile and almost non-existent (about 1%) among households in the highest income tertile (Table 2, Fig 2a).

### Association between FPC coverage and food insufficiency by income, race, and ethnicity

High FPC coverage was associated with small, significantly lower prevalence of food insufficiency during the pandemic among non-Hispanic Black and white households in the lower third income categories compared to similar households

**Table 1. Characteristics of Census Household Pulse Survey respondents and states included in an analysis of food sufficiency during the COVID-19 pandemic (March 2020 – May 2023) by food policy council coverage.**

| Individual Level Variables | Low FPC Coverage (<=15%) (N = 772,754) | High FPC Coverage (>15%) (N = 1,136,893) |
|---|---|---|
| Age in years, mean (sd) | 53.9 (15.6) | 53.4 (15.7) |
| Gender, n (%) | | |
| Male | 317100 (41.0) | 478126 (42.1) |
| Female | 455654 (59.0) | 658767 (57.9) |
| Ethnicity and race, n (%) | | |
| Hispanic | 58772 (7.6) | 99740 (8.7) |
| White, non-Hispanic | 606599 (78.5) | 863924 (76.0) |
| Black, non-Hispanic | 51102 (6.6) | 72452 (6.4) |
| Asian, non-Hispanic | 26764 (3.5) | 60076 (5.3) |
| Other, non-Hispanic | 29517 (3.8) | 40701 (3.6) |
| Education level, n (%) | | |
| Less than high school | 12834 (1.7) | 19476 (1.7) |
| High school or equivalent | 90628 (11.7) | 115859 (10.2) |
| More than high school | 669292 (86.6) | 1001558 (88.1) |
| Total household income level, n (%) | | |
| Lower third | 226963 (29.4) | 309205 (27.2) |
| Middle third | 255404 (33.0) | 357817 (31.5) |
| Upper third | 290387 (37.6) | 469871 (41.3) |
| Number of non-adults, n (%) | | |
| 0 | 520791 (67.4) | 774533 (68.1) |
| 1 or 2 | 199729 (25.8) | 294258 (25.9) |
| 3 or more | 52234 (6.8) | 68102 (6.0) |
| **State Level Variables** | **Low FPC Coverage (<=15%) (N = 25)** | **High FPC Coverage (>15%) (N = 25)** |
| Region | | |
| Midwest | 4 (16.0) | 8 (32.0) |
| Northeast | 5 (20.0) | 4 (16.0) |
| South | 10 (40.0) | 6 (24.0) |
| West | 6 (24.0) | 7 (28.0) |
| Presence of state-level council, n (%) | 18 (72.0) | 21 (84.0) |
| Percent of population living in rural area, mean (sd) | 30 (16.2) | 25.1 (13.1) |
| Median household income, mean (sd) | 69040.3 (12941.3) | 73572.1 (10233.2) |
| Percent of eligible individuals participating in SNAP, mean (sd) | 82.8 (11.1) | 83.4 (9.5) |

in states with low FPC coverage (Fig 2c, 2d). Food insufficiency among lower income non-Hispanic Black households (<$35K/year) was 26.03% in states with low FPC coverage and 24.73% in states with high coverage (prevalence ratio 1.05, 95% CI 1.02–1.09, p = 0.003). It was 12.68% in states with low FPC coverage and 12.42% in states with high coverage (prevalence ratio 1.02, 95% CI 1.00–1.04, p = 0.01) among lower income white non-Hispanic households (<$50K/year) (**Table 2**). There was no difference in food insufficiency in the middle income tertiles but there was a significant

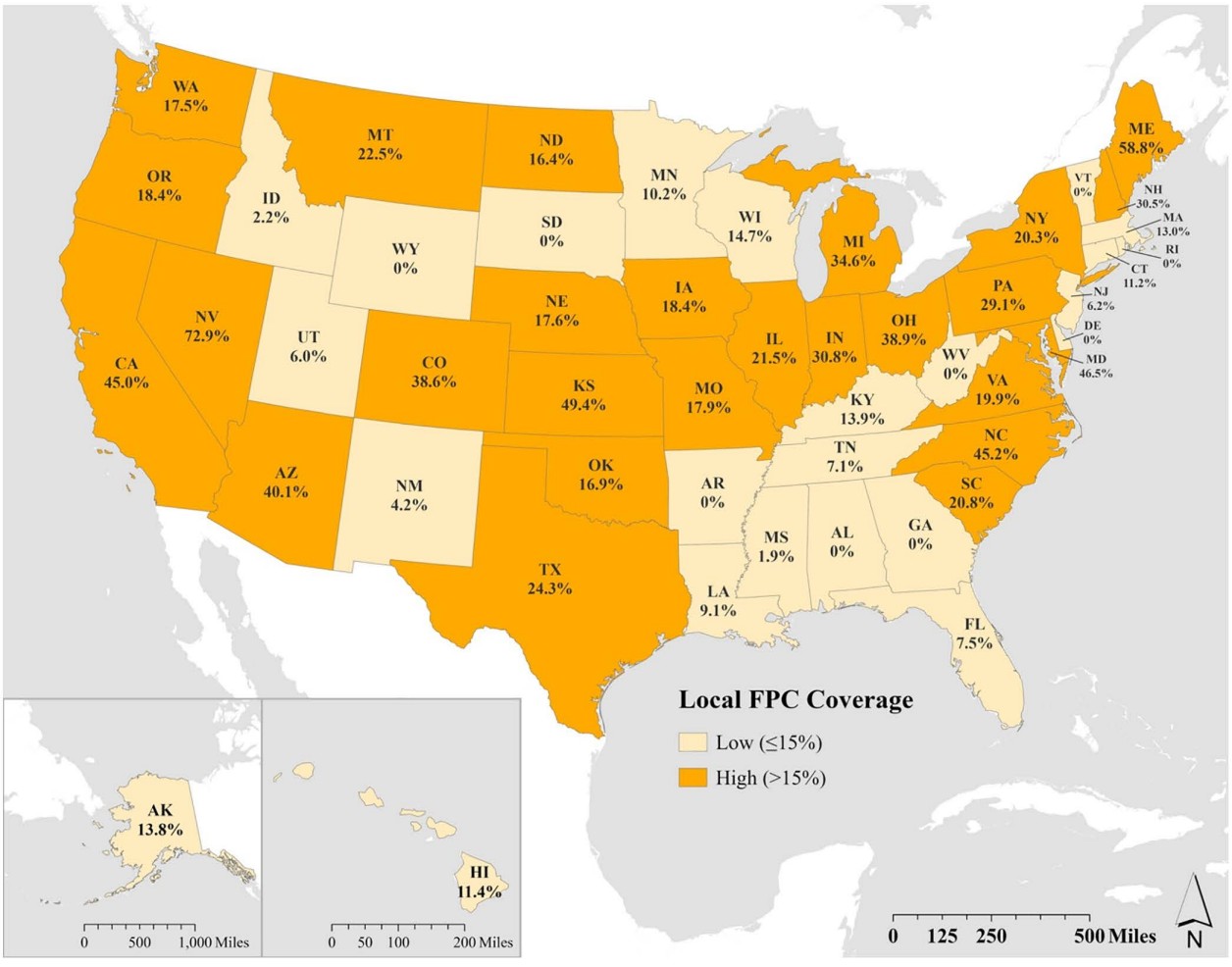

**Fig 1. Map of the United States a) showing the percent of a state's population that resides in an area where there was an active, local food policy council (FPC) in 2021 and b) showing states categorized as having low (<15%) or high (>15%) FPC coverage.**

difference in food insufficiency among higher income Black households in states with low and high FPC coverage (3.55% insufficiency in low FPC states compared to 3.01% in high FPC states).

FPC coverage was not significantly associated with food insufficiency during the COVID-19 pandemic among Hispanic, Asian, and Other non-Hispanic households (Fig 2a,2e,2f). There were significant differences in food insufficiency among the highest tertile Asian and Other non-Hispanic households; however, food insufficiency levels were very low among those groups (Table 2). Quarterly prevalence ratios and 95% CIs of food insufficiency reported on the US Census HPS between May 2020 and May 2023 comparing states with low (≤15%) and high (>15%) active local FPC coverage by race and ethnic-specific household income tertiles are available in S1 Fig.

## Discussion

To our knowledge, this is the first study to empirically assess associations between local FPC coverage and food insufficiency and to stratify associations by income, race, and ethnicity. We hypothesized that the presence of FPCs could mitigate the effects of the pandemic due to FPCs' efforts to strengthen food systems through

**Table 2. Prevalence of food insufficiency in communities with low (<15%) and high (>15%) local food policy council coverage by ethnicity, race, and household income level.**

| | Sample | Prevalence (%) | Prevalence (%) | Prevalence Ratio | P-value |
|---|---|---|---|---|---|
| | Size | Low Coverage | High Coverage | (95% CI) | |
| Overall aggregate | 1,909,647 | 6.03 | 5.62 | 1.07 (1.06–1.08) | < 0.001 |
| Overall | | | | | |
| Lower Third | 536,168 | 15.13 | 14.38 | 1.05 (1.04–1.07) | < 0.001 |
| Middle Third | 613,221 | 4.52 | 4.48 | 1.01 (0.99–1.03) | 0.36 |
| Upper Third | 760,258 | 1.09 | 1 | 1.09 (1.04–1.14) | < 0.001 |
| Hispanic | | | | | |
| Lower Third (<$35K) | 45,359 | 25.11 | 24.95 | 1.01 (0.97–1.04) | 0.71 |
| Middle Third ($35K – $75K) | 50,303 | 11.18 | 11.14 | 1.00 (0.95–1.06) | 0.88 |
| Upper Third (> $75K) | 62,850 | 3.02 | 2.98 | 1.01 (0.92–1.11) | 0.8 |
| Black non-Hispanic | | | | | |
| Lower Third (<$35K) | 41,264 | 26.03 | 24.73 | 1.05 (1.02–1.09) | 0.003 |
| Middle Third ($35K – $75K) | 40,204 | 12.62 | 12.53 | 1.01 (0.96–1.06) | 0.79 |
| Upper Third (> $75K) | 42,086 | 3.55 | 3.01 | 1.18 (1.06–1.31) | 0.002 |
| White non-Hispanic | | | | | |
| Lower Third (<$50K) | 400,463 | 12.68 | 12.42 | 1.02 (1.00–1.04) | 0.01 |
| Middle Third ($50K – $100K) | 473,436 | 3.02 | 3.01 | 1.00 (0.97–1.03) | 0.93 |
| Upper Third (> $100K) | 596,624 | 0.65 | 0.62 | 1.06 (1.00–1.13) | 0.06 |
| Asian non-Hispanic | | | | | |
| Lower Third (<$75K) | 29,635 | 8.51 | 8.54 | 1.00 (0.91–1.08) | 0.94 |
| Middle Third ($75K – $150K) | 28,307 | 1.55 | 1.58 | 0.98 (0.79–1.22) | 0.88 |
| Upper Third (> $150K) | 28,898 | 0.11 | 0.39 | 0.28 (0.17–0.42) | < 0.001 |
| Other non-Hispanic | | | | | |
| Lower Third (<$35K) | 19,447 | 27.92 | 26.7 | 1.05 (1.00–1.09) | 0.059 |
| Middle Third ($35K – $75K) | 20,971 | 12.24 | 12 | 1.02 (0.95–1.10) | 0.6 |
| Upper Third (> $75K) | 29,800 | 3.67 | 2.81 | 1.30 (1.15–1.48) | < 0.001 |

All estimates were adjusted for age and gender of survey completer, and percentage of population living in a rural area.

programs, policies, and relationships across sectors under normal and crisis conditions [28–30]. Our results suggest that local FPCs are well positioned to rapidly respond to food system-related shocks, and their actions could be especially impactful for vulnerable populations that have fewer resources and less capacity to manage disruptive shocks.

Income, race, and ethnicity moderated the effect of FPC coverage on household food insufficiency in our analysis. We found that households from low income and racial minorities were at higher risk of experiencing food insufficiency during the pandemic, and that FPCs' efforts to reduce food insufficiency for households at risk may have been especially impactful for those groups. Decades of survey data indicate that food insufficiency is relatively common among low-income families and almost non-existent among households earning >185% of the poverty level, though household assets and area-specific cost of living are also important determinants of food insufficiency [31,32]. Income distribution was clearly different across racial and ethnic groups, reflecting known economic disparities in the US [33,34]. Before the pandemic, food insecurity rates for Black and Hispanic households were 21.2% and 16.2% compared to the national average of 11.1% [35]. Our results showed persistent disparities in food insufficiency (a different concept than food insecurity) among these groups during the pandemic.

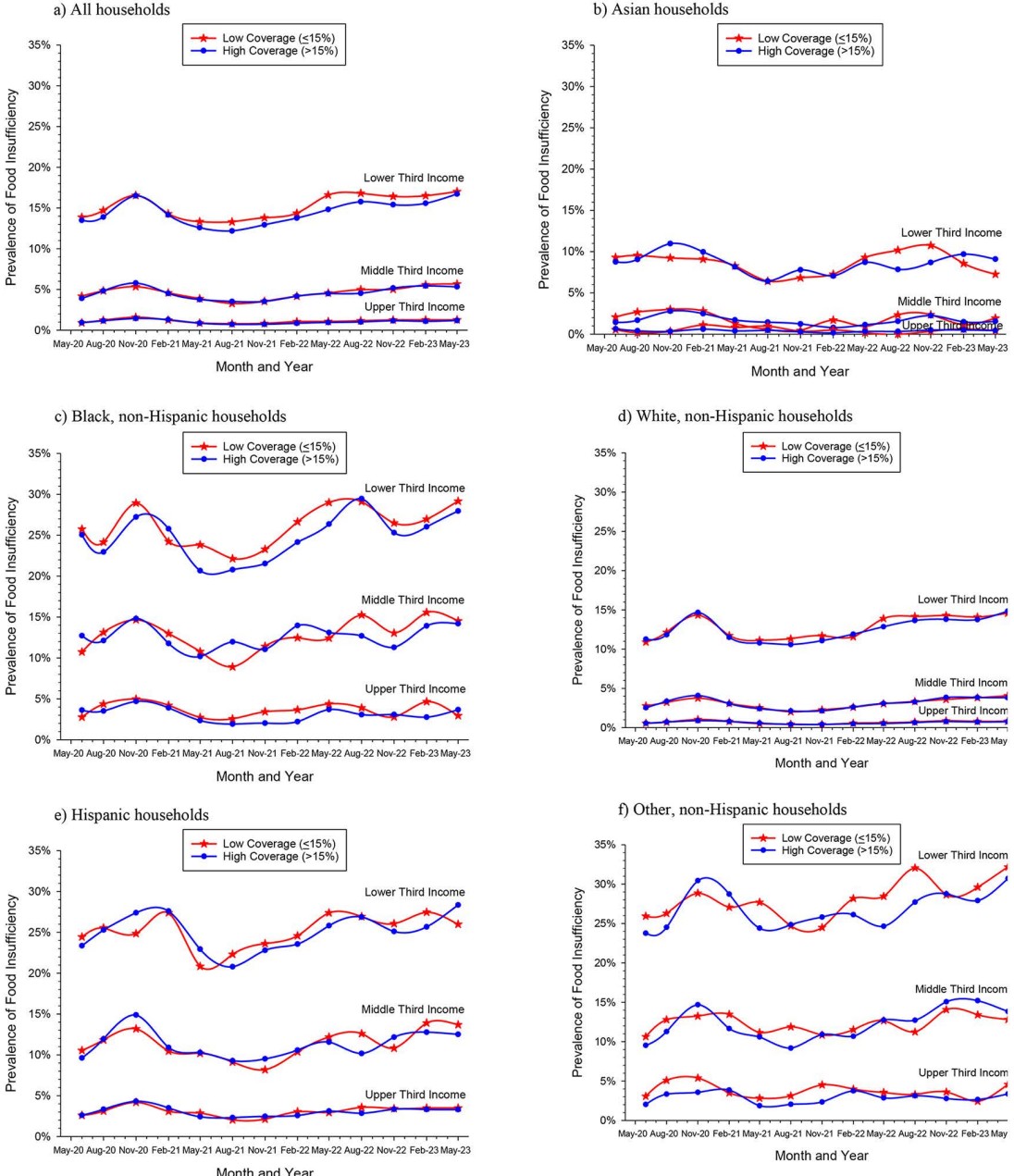

**Fig 2. Quarterly prevalence of food insufficiency reported on the US Census Household Pulse Survey between May 2020 and May 2023 in states with low (<15%) and high (>15%) active local food policy council coverage by race and ethnic-specific household income tertiles.** (a) in all households, (b) Asian non-Hispanic households, (c) Black non-Hispanic households, (d) white non-Hispanic households, (e) Hispanic households, and (f) other non-Hispanic households.

During the timeframe of our analysis, several Economic Impact Payments (i.e., "stimulus checks") of up to $600–1,400 per adult and $600 per qualifying child were sent to families as emergency measures to boost household income and buoy the national economy during the public health crisis [36]. Federal nutrition safety net measures, such as increasing Supplemental Nutrition Assistance Program (SNAP) benefits, creating Pandemic Electronic Benefits Transfer (P-EBT),

allowing SNAP benefits to be used for online grocery purchases, and other programs, provided additional support to low-income households, especially those with children [27]. We found that households in states with more local FPC coverage were less likely to experience food sufficiency, within the context of these and other federal actions and benefits. More research is needed to determine whether pandemic relief policy supports diminished or amplified the association between FPC coverage and household food insufficiency.

It is very difficult to trace "upstream" efforts like assembling and operating a local FPC and population-level health effects because there are so many steps and delays along a causal pathway connecting the two, and population-level health outcomes are influenced by a multitude of factors beyond those that local FPCs can impact [16,37]. Successful examples of tracing cross-sector collaboration prevalence to population-level health outcomes include reductions in preventable death rates and reducing risky behaviors among adolescents [38,39]. In those instances, and in this study, a sufficiently large sample size is needed to detect associations in a "noisy" environment with many policies, programs, and economic shifts occurring over the study period. Evidence linking upstream efforts to downstream outcomes, like the reports from the FPN project at the Johns Hopkins Center for a Livable Future and other groups that documented the actions FPCs took during the pandemic [15], is crucial for understanding how complex community-level change can unfold.

Our study has limitations. We used the FPN's 2021 census of FPCs to determine FPC coverage; if FPCs exist that are not included in the census we may have undercounted FPC coverage. We accounted for state-level FPCs in our analysis, but we did not explore the role of state-level FPCs or conveners in this study. Further research should investigate how state-level FPCs and groups influence the effectiveness of local FPCs. The main outcome was food insufficiency, which is not equivalent to the more comprehensive and commonly used concept of food security [22]. Our sample contained a higher percentage of greater than a high school degree than the national average (85% in our sample compared to 63.2% nationally [40]). Completing at least some college is associated with less food insecurity after controlling for poverty, suggesting our sample may be less susceptible to food insufficiency than the general population [41]. We conducted the analysis during COVID-19, which created an opportunity to compare food insufficiency during a time when many FPCs were actively working to address food insufficiency in their communities [15] Future studies should analyze FPCs' impact under a variety of conditions. For example, scientists in Maryland are studying FPCs' potential buffering effects for other types of shocks, such as climate-related change and social unrest [42]. This study was a natural experiment, so there could be unmeasured confounding factors, such as more effort to address food insufficiency in states that happen to have higher FPC coverage, independent of FPCs' activities. For example, SNAP recipients in some states could use their benefits to purchase groceries online by May 2020, while online shopping implementation took much longer in other states [43]. In states like Maryland, where state agencies convened local food system groups, such as councils, such coordination may have led to better food security metrics. We could not account for the broad heterogeneity in how states responded to food insufficiency during the pandemic in this analysis. Finally, we only studied the effect of FPC coverage on state-level food insufficiency during the pandemic as we did not capture program, policies, or other groups that may also be targeting food insufficiency.

## Conclusion

A higher prevalence of local FPCs in a state was associated with lower prevalence of household food insufficiency during the COVID-19 pandemic among low-income and minoritized households. Amidst calls for strategies that address social determinants of health such as food access that promote health equity and community resilience, local FPCs should be strongly considered for further investment and study.

## Supporting information

**S1 Table. U.S. Census Household Pulse Survey data collection cycle dates and corresponding time point in an analysis of household food insufficiency during the COVID-19 public health from May 2020–20203.**
(DOCX)

**S2 Table. Characteristics comparison between analytical and exclusion sample of Census Household Pulse Survey respondents included in an analysis of food sufficiency during the COVID-19 pandemic (March 2020 – May 2023) by food policy council coverage.**
(DOCX)

**S1 Fig. Quarterly prevalence ratios and 95% confidence intervals (95% CIs) of food insufficiency reported on the US Census Household Pulse Survey between May 2020 and May 2023 comparing states with low (<15%) and high (>15%) active local food policy council coverage by race and ethnic-specific household income tertiles.** (a) in all households, (b) Asian non-Hispanic households, (c) Black non-Hispanic households, (d) white non-Hispanic households, (e) Hispanic households, and (f) other non-Hispanic households. Dotted line represents prevalence ratio of 1.0 (i.e., no difference between food insufficiency prevalence among households in states with low compared to high active local food policy council coverage).
(DOCX)

## Author contributions

**Conceptualization:** Larissa Calancie.

**Data curation:** Larissa Calancie, Yongyi Pan, Karen Bassarab, Anne Palmer, Misha Eliasziw.

**Formal analysis:** Larissa Calancie, Yongyi Pan, Misha Eliasziw.

**Funding acquisition:** Larissa Calancie.

**Investigation:** Larissa Calancie, Kristen Cooksey Stowers, Misha Eliasziw.

**Methodology:** Larissa Calancie, Misha Eliasziw.

**Project administration:** Larissa Calancie.

**Resources:** Larissa Calancie, Karen Bassarab, Anne Palmer.

**Supervision:** Larissa Calancie.

**Visualization:** Larissa Calancie, Yongyi Pan, Misha Eliasziw.

**Writing – original draft:** Larissa Calancie, Yongyi Pan, Karen Bassarab, Kristen Cooksey Stowers, Anne Palmer, Misha Eliasziw.

**Writing – review & editing:** Larissa Calancie, Karen Bassarab, Kristen Cooksey Stowers, Anne Palmer, Misha Eliasziw.

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
