## [Decision Letter · Decision Letter 0]

17 Nov 2025

Dear Dr. Calancie,

Thank you for submitting your manuscript to PLOS ONE. After careful consideration, we feel that it has merit but does not fully meet PLOS ONE’s publication criteria as it currently stands. Therefore, we invite you to submit a revised version of the manuscript that addresses the points raised during the review process.

We look forward to receiving your revised manuscript.

Kind regards,

Hao Wang

Academic Editor

PLOS ONE

Journal Requirements:

“L.C. received funding from the Kansas Health Foundation

Authors of this publication are members of the Food Policy Council Working Group, part of the Nutrition and Obesity Policy Research and Evaluation Network (NOPREN). NOPREN is supported by Cooperative Agreement Number 5U48DP00498-05, funded by the Centers for Disease Control and Prevention’s (CDC) Division of Nutrition, Physical Activity, and Obesity (DNPAO) and Prevention Research Centers Program. The findings and conclusions in this publication are those of the author(s) and do not necessarily represent the official position of the CDC or the U.S. Department of Health and Human Services (DHHS).”

3. Thank you for uploading your study's underlying data set. Unfortunately, the repository you have noted in your Data Availability statement does not qualify as an acceptable data repository according to PLOS's standards.

4. We note that Figure 1 in your submission contain map images which may be copyrighted. All PLOS content is published under the Creative Commons Attribution License (CC BY 4.0), which means that the manuscript, images, and Supporting Information files will be freely available online, and any third party is permitted to access, download, copy, distribute, and use these materials in any way, even commercially, with proper attribution. For these reasons, we cannot publish previously copyrighted maps or satellite images created using proprietary data, such as Google software (Google Maps, Street View, and Earth). For more information, see our copyright guidelines: http://journals.plos.org/plosone/s/licenses-and-copyright.

Reviewer's Responses to Questions

**Comments to the Author**

1. Is the manuscript technically sound, and do the data support the conclusions?

Reviewer #1: Partly

Reviewer #2: Yes

2. Has the statistical analysis been performed appropriately and rigorously?

Reviewer #1: Yes

Reviewer #2: Yes

3. Have the authors made all data underlying the findings in their manuscript fully available?

Reviewer #1: Yes

Reviewer #2: Yes

4. Is the manuscript presented in an intelligible fashion and written in standard English?

Reviewer #1: Yes

Reviewer #2: Yes

Reviewer #1: Thank you for the opportunity to review this interesting study “Association Between Local Food Policy Council Coverage and Longitudinal Household Food Insufficiency During COVID-19, Stratified by Race, Ethnicity, and Income”.

This is a retrospective, descriptive study that is described by the authors as a natural experiment due to food insufficiency during a crisis (COVID-19 pandemic). The aims of this study were to 1.) determine whether households living in states with higher local FPC coverage were less likely to experience food insufficiency during the COVID-19 pandemic compared to households in states with lower local FPC coverage, and 2.) whether the association between local FPC coverage and food insufficiency were moderated by race, ethnicity, and income.

Food insufficiency was defined as a response to the Household Pulse Survey (HPS) Question of “In the last 7 days, which of these statements best describes the food eaten in your household?” with an answer of either 3.) sometimes not enough to eat, or 4.) Often not enough to eat. Local FPC coverage was defined as the percentage of a state population living in an area with an active FPC that operated locally. State-level councils were not counted towards local FPC coverage. Individual and household level variables of age, gender, race and ethnicity, household income and education level were extracted from HPS data.

The initial HPS sample included 4,412,510 participants aged 17 years or older. Participants were excluded who did not have a record of food insufficiency (n = 341,785), record of income (n = 518,227) and participants from the District of Columbia (n = 52,062). These exclusion criteria resulted in 3,500,436 participants.

In this study, authors found that households in states where more than 15% of their population lived in an area with an active FPC during the COVID-19 pandemic were less likely to report food insufficiency than households in states with lower FPC coverage. They also found several moderating effects based on income, race, and ethnicity.

The authors described multiple limitations to the study, including the difficulty of linking “upstream” efforts like operating an a local FPC to population-level effects due to the presence of confounders.

This is an interesting study. Strengths of this study include the robust, publicly available data and number of participants, the stratification based on individual and household factors, and well as the novelty of the topic. However, there are some major issues that affect the quality of this study including addressing confounding and interpretation of data.

Abstract

1. Well written and follows the guidelines

Introduction

1. Overall, well written and describes the rationale behind the potential association of FPC coverage and food insufficiency as well as using the COVID-19 pandemic as a natural experiment.

Methods

1. Line 140, change percent to percentage

2. FPC coverage is based on 2021 data but outcome data spans a wider range, it may be appropriate to use FPC coverage for each year

3. FPC coverage is dichotomous at the level of 15% coverage, with the explanation that this splits the number of states in half. Is this method appropriate given the wide range of FPC coverage within each group? (0-15% and 16%+) Would it be more appropriate to analyze this data using a continuous model?

4. Income is organized into tertiles, but not based on income levels by state; cost of living varies greatly across this geographic distribution so it may be more appropriate to organize these data with respect to state level income distributions

5. A high number of participants are excluded due to no record of income. This should be commented on in greater detail. This excluded group may represent a distinct subset of the total population (such as low income or minority race/ethnicity).

Results

1. Many of the results are reported as statistically significant but represent a numerically small difference. The authors could comment on the practical significance of the difference, especially with respect to policy implications

Discussion

1. There is a high percentage of greater than high school education in this data, this should be commented on in more detail

2. Conclusions should be described as an association instead of a potential effect or causal factor.

3. More discussion should be dedicated to state level confounders including heterogenous state-level COVID response.

Reviewer #2: Lines 68-71 This sentence is structurally difficult to read. I think if the in parenthesis examples were removed from their current placement and included at the end of the sentence that it would be clearer for the reader. "..for similar and policy specific reasons such as (insert examples)."

Line 237 This sentence seems to contradict itself with higher percentage in lower coverage. The text immediate preceding the percentage in parentheses says the percentage was lower in the low coverage states.

Line 291-295 Please consider breaking into two separate sentences for easier readability.

**Do you want your identity to be public for this peer review?** For information about this choice, including consent withdrawal, please see our Privacy Policy

Reviewer #1: No

Reviewer #2: No

---

## [Author Response · Author response to Decision Letter 1]

11 Feb 2026

FPC paper – Response to reviewers

Editor comments:

Thank you, the revised manuscript adheres to PLOS ONE’s style requirements now.

Thank you, we included the statement above after our financial disclosures.

3. Thank you for uploading your study's underlying data set. Unfortunately, the repository you have noted in your Data Availability statement does not qualify as an acceptable data repository according to PLOS's standards.

We checked the recommended repositories and Harvard Dataverse Network is on the list of recommended options. Our data is published there. I used the Harvard Dataverse Network link from the PLOS ONE link pasted above and searched our study to double check and indeed, it is all our data is available on that repository. We provided the relevant DOI and related data information.

4. We note that Figure 1 in your submission contain map images which may be copyrighted.

Thank you, we created a new figure using ArcGIS and acknowledged the data source documentation on the figure and in the methods section (“All spatial processing and map visualizations were conducted in ArcGIS Pro (version 3.5.3, Esri Inc, 2025, Redlands, CA). US state boundary shapefiles were obtained from the US Census Bureau’s TIGER/Line dataset.)”

We added captions for supplemental files to the end of the manuscript and followed supplemental material naming conventions.

The reviewers did not recommend citing specific previously published works.

Reviewer #1

Abstract

1. Well written and follows the guidelines

Thank you

Introduction

1. Overall, well written and describes the rationale behind the potential association of FPC coverage and food insufficiency as well as using the COVID-19 pandemic as a natural experiment.

Thank you

Methods

1. Line 140, change percent to percentage

Done

2. FPC coverage is based on 2021 data but outcome data spans a wider range, it may be appropriate to use FPC coverage for each year

The Food Policy Network surveys FPCs every 2 years and there are few changes in the number of councils each year. There is also a delay between when the survey is fielded and when the results are available. Therefore, we are confident that the 2021 data is a strong (and feasible) representation of FPC coverage across the food insufficiency data range.

3. FPC coverage is dichotomous at the level of 15% coverage, with the explanation that this splits the number of states in half. Is this method appropriate given the wide range of FPC coverage within each group? (0-15% and 16%+) Would it be more appropriate to analyze this data using a continuous model?

Council coverage was dichotomized at 15% to ease the interpretation of the regression results as this factor was included in the model in a three-way cross-product term along with quarter and income level. Had council coverage been analyzed as a continuous variable, the results would have been reported as regression beta coefficients rather prevalence and prevalence ratios, which we believe are more meaningful.

4. Income is organized into tertiles, but not based on income levels by state; cost of living varies greatly across this geographic distribution so it may be more appropriate to organize these data with respect to state level income distributions

We agree that cost of living varies from state to state as well as many other factors that affect a person’s socioeconomic status, including within-state cost of living variation and racial/ethnic identity. Early on in our analyses we recognized that the dollar value corresponding to the lower, middle, and upper tertile of household income varied by racial/ethnic identity and therefore different tertile cut points were assigned to recognize the income disparity.

5. A high number of participants are excluded due to no record of income. This should be commented on in greater detail. This excluded group may represent a distinct subset of the total population (such as low income or minority race/ethnicity).

We compared the excluded group to the included group and reported the findings in a new table (Suppl Table 2). We summarized the differences in the results section (“More Hispanic, Black, and Asian households were excluded due to missing household income data compared to white households and more households with high school or less education were excluded than households with higher education levels) and commented on the differences in the limitations section of the discussion (“Households that were excluded due to missing household income data were more likely to be racial or ethnic minorities and report lower levels of education; such factors are associated with higher risk of food insufficiency, so we may have underestimated food insufficiency in our analysis.”)

Results

1. Many of the results are reported as statistically significant but represent a numerically small difference. The authors could comment on the practical significance of the difference, especially with respect to policy implications

Good point. We revised the sentence reporting our main result to read: “Overall, slightly fewer households in states where more than 15% of their population lived in an area with an active FPC reported food insufficiency during the COVID-19 pandemic than households in states with lower FPC coverage according to multivariate regression modeling.”

We also added “small” to this sentence: “High FPC coverage was associated with small, significantly lower prevalence of food insufficiency during the pandemic among non-Hispanic Black and white households in the lower third income categories compared to similar households in states with low FPC coverage (Figure 2c,d).”

We also added a sentence to the end of the paragraph about policy supports during the pandemic that reads: “More research is needed to determine whether pandemic relief policy supports diminished or amplified the association between FPC coverage and household food insufficiency.”

Discussion

1. There is a high percentage of greater than high school education in this data, this should be commented on in more detail

Good point. We added the following sentences to the limitations paragraph in the discussion: “Our sample contained a higher percentage of greater than a high school degree than the national average (85% in our sample compared to 63.2% nationally (40). Completing at least some college is associated with less food insecurity after controlling for poverty, suggesting our sample may be less susceptible to food insufficiency than the general population (41).”

2. Conclusions should be described as an association instead of a potential effect or causal factor.

We modified the first sentence of the conclusion to read: “A higher prevalence of local FPCs in a state was associated with lower prevalence of household food insufficiency during the COVID-19 pandemic among low-income and minoritized households.”

3. More discussion should be dedicated to state level confounders including heterogenous state-level COVID response.

Thank you, we added the following sentences to the limitations paragraph in the discussion: “For example, SNAP recipients in some states could use their benefits to purchase groceries online by May 2020 while online shopping implementation took much longer in other states (43). We could not account for the broad heterogeneity in how states responded to food insufficiency during the pandemic in this analysis.”

Reviewer #2:

Lines 68-71 This sentence is structurally difficult to read. I think if the in parenthesis examples were removed from their current placement and included at the end of the sentence that it would be clearer for the reader. "..for similar and policy specific reasons such as (insert examples)."

Thank you, we revised that sentence based on your recommendation.

Line 237 This sentence seems to contradict itself with higher percentage in lower coverage. The text immediate preceding the percentage in parentheses says the percentage was lower in the low coverage states.

Thank you, we changed that sentence to read: “Individual and state level characteristics were similar in both groups of states.”

Line 291-295 Please consider breaking into two separate sentences for easier readability.

We broke it into two sentences.

---

## [Editor Report · Decision Letter 1]

9 Mar 2026

Association Between Local Food Policy Council Coverage and Longitudinal Household Food Insufficiency During COVID-19, Stratified by Race, Ethnicity, and Income

PONE-D-25-54719R1

Dear Dr. Calancie,

We’re pleased to inform you that your manuscript has been judged scientifically suitable for publication and will be formally accepted for publication once it meets all outstanding technical requirements.

Kind regards,

Hao Wang

Academic Editor

PLOS One
---

## [Editor Report · Acceptance letter]

PONE-D-25-54719R1

PLOS One

Dear Dr. Calancie,

I'm pleased to inform you that your manuscript has been deemed suitable for publication in PLOS One. Congratulations! Your manuscript is now being handed over to our production team.

Kind regards,

on behalf of

Dr. Hao Wang

Academic Editor

PLOS One